# Adherence to the Healthy Eating Index-2015 across Generations Is Associated with Birth Outcomes and Weight Status at Age 5 in the Lifeways Cross-Generation Cohort Study

**DOI:** 10.3390/nu11040928

**Published:** 2019-04-25

**Authors:** Pilar Navarro, John Mehegan, Celine M. Murrin, Cecily C. Kelleher, Catherine M. Phillips

**Affiliations:** HRB Centre for Diet and Health Research, School of Public Health, Physiotherapy, and Sports Science, University College Dublin, Dublin 4, Ireland; pilar.navarro@ucd.ie (P.N.); john.mehegan@ucd.ie (J.M.); celine.murrin@ucd.ie (C.M.M.); cecily.kelleher@ucd.ie (C.C.K.)

**Keywords:** healthy eating index, diet quality, dietary patters, weight status, birth outcomes, parental lineages

## Abstract

Maternal dietary quality during pregnancy is associated with offspring outcomes. These associations have not been examined in three-generation families. We investigated associations between parental and grandparental dietary quality, determined by healthy eating index (HEI)-2015, and offspring birth outcomes and weight status at age 5. The Lifeways cohort study in the Republic of Ireland comprises 1082 index-child’s mothers, 333 index-child’s fathers, and 707 grandparents. HEI-2015 scores were generated for all adults from prenatal dietary information collected using a validated food frequency questionnaire. In an adjusted model, greater adherence to the maternal HEI was associated with lower likelihood of low birth weight (LBW) (OR: 0.72, 95% CI: 0.50–0.99, *p* = 0.04). Similarly, maternal grandmothers (MGM) with higher HEI scores were less likely to have grandchildren with LBW (OR: 0.87, 95% CI: 0.61–0.96, *p* = 0.04) and more likely to have macrosomia (OR: 1.10, 95% CI: 1.01–1.22, *p* = 0.03). Higher paternal and paternal grandmothers (PGM) HEI scores were associated with lower likelihood of childhood obesity (OR: 0.89, 95% CI: 0.30–0.94, *p* = 0.03) and overweight (OR: 0.83, 95% CI: 0.22–0.99, *p* = 0.04), respectively. Mediation analysis showed significant direct relationship of MGM and PGM HEI scores on grandchildren’s birthweight and obesity, respectively. In conclusion, maternal line dietary quality appears to influence fetal growth whereas paternal line dietary quality appears to influence postnatal growth.

## 1. Introduction

Maternal diet plays an important role in determining birth outcomes and offspring health [1,2]. However, the specific dietary requirements for optimal fetal growth and development remain unknown. Familial associations may influence offspring growth and development outcomes through shared lifestyle and social environments [3], including diet, heritable genetic and epigenetic mechanisms [4], and for maternal lines in particular, intra-uterine and mitochondrial pathways [5]. According to the developmental origins of health and disease (DOHaD) hypothesis, transient early life exposures, including intrauterine nutrition, during critical periods of development, such as pregnancy, may alter normal physiology affecting offspring health in later life [3]. Importantly such long-term consequences may not be limited to one generation but may lead to adverse health outcomes in future generations even in the absence of the exposure, and these patterns may contrast in maternal and paternal lines [6,7]. Thus suboptimal nutrition during pregnancy may perpetuate intergenerational transmission of adverse health outcomes.

Studies investigating maternal diet have mainly focused on adequate intake of selected macronutrients and micronutrients during pregnancy [8,9,10,11,12]. In family cohorts, including the Lifeways study, significant positive three-generation correlations were observed for nutrient intakes of maternal grandmother–mother–child triads, but not found in paternal lines [5].

Several studies have examined the associations between overall maternal diet quality and offspring outcomes [13,14,15,16,17,18,19], through different index scores such as the healthy eating index (HEI) [13], the alternate healthy eating index (AHEI) [14,16], adherence to Mediterranean diet score during pregnancy [17,18], and the dietary approaches to stop hypertension (DASH) score [19].

The HEI-2015 is the latest update of a diet quality index designed by the United States Department of Agriculture [20,21,22]. Several studies have described associations of maternal HEI scores with birth outcomes [23,24,25]. Thus far, the focus has been on maternal HEI scores and the relationship between father–child diet quality has been less frequently examined, although a few studies found a significant positive father–child relationship [26,27]. To our knowledge, no study has investigated the influence of diet quality across three generations and their potential influence on offspring outcomes. Thus, our objective was to investigate any associations between maternal, paternal, and grandparental dietary quality, determined by HEI-2015 scores, and offspring birth outcomes and weight status at 5 years in the Lifeways cross-generation cohort study in the Republic of Ireland.

## 2. Materials and Methods

### 2.1. Participants

The Lifeways cross generation cohort study is a prospective family study that has been described in detail elsewhere [28,29]. The study objectives were to document health status, diet, and lifestyle in the family members and to establish patterns and links across generations. Briefly, Irish-born mothers were recruited by a midwife during their first antenatal booking visit in two maternity hospitals in the Republic of Ireland between 2001–2003. A cohort of 1094 live infants were born to 1082 recruited mothers, which formed the eligible participant pool for the current study. A total of 585 children were assessed at 5 years. The participating mothers’ partner and a least one of the index-child grandparents (maternal grandmother (MGM), maternal grandfather (MGF), paternal grandmother (PGM), and paternal grandfather (PGF)) were directly contacted by the Lifeways research team (participant mothers having given their contact details) and invited to participate also. If they agreed (signed the consent form), they then returned baseline self-completed questionnaires. Three hundred and thirty-three index-child’s fathers and 707 of any four grandparents did so. A flow chart outlining the study participants included in the current analysis is presented in Appendix A.

### 2.2. Ethics

Ethical approval was granted by ethical committees of the Coombe University Hospital, Dublin, University College Dublin, Irish College of General Practitioners and University College Hospital, Galway, Ireland. Written informed consent was collected from all women upon recruitment and at all subsequent sweeps of the study.

### 2.3. Dietary Intake Assessment

All adults completed the same questionnaire. Habitual dietary intakes of the women during the first trimester of pregnancy and 5 years postnatally were assessed by a validated 149-item semi quantitative food frequency questionnaire (FFQ), which has been validated for use in the Irish population [11]. Participants were asked about their average consumption frequency (9 levels, from ‘never or less than once per month’ to ‘6+ per day’) of each food items during the first 12–16 weeks of pregnancy. The daily quantities of food intakes were then derived by multiplying the frequencies per day with standard portion sizes [30]. The FFQ assessed dietary supplements use. Daily energy and nutrient intakes were computed for each participant using an in-house software program (FFQ Software Ver 1.0; developed by the National Nutrition Surveillance Centre, School of Public Health, Physiotherapy and Sports Science, University College Dublin, Belfield, Dublin 4, Ireland) based on the McCance and Widdowson food tables [30].

### 2.4. HEI-2015 Scoring

The HEI-2015 is a measure of overall diet quality that measures alignment with the 2015–2020 dietary guidelines for Americans [31]. The HEI-2015 contains 13 components which are scored on a density basis out of 1000 calories, with the exception of fatty acids, which is a ratio of unsaturated to saturated fatty acids (SFAs) [20,22]. Total fruits, whole fruits, total vegetables, greens and beans, total protein containing foods, and seafood and plant proteins scored 5 in the highest consumption and 0 in the lowest consumption. The highest consumption of three components including whole grains, dairy, and fatty acids (ratio of poly- and monounsaturated fatty acids (PUFAs and MUFAs) to SFAs) scored as 10 and the lowest consumption scored as 0. Four components (refined grains, sodium, added sugars, and saturated fats) scored 10 in the lowest consumption and 0 in the highest consumption [22,32]. Component scores are summed to yield a total score ranging from 0 to 100, with a higher score indicating greater adherence to the dietary guidelines for Americans.

### 2.5. Offspring Outcomes Assessment

Information on birth outcomes and infant gender were abstracted from linked hospital records. Adverse birth outcomes: (1) Low birth weight (LBW), BW < 2500 g; (2) macrosomia, BW > 4000 g; (3) pre-term birth, delivery before 37 completed weeks of gestation; and (4) post-term birth, delivery at and after 42 completed weeks of gestation) were defined based on standard clinical cut-offs [33,34,35]. At a follow-up home visit when the children were 5 years old, their weights and heights were measured to the nearest 0.1 kg and 0.1 cm, respectively, using standardized protocols by trained research personnel; instruments used were SECA digital weighing scale and portable Leicester height scale, both purchased and calibrated from Chasmors Ltd., London, UK. BMI was derived using the formula weight/height^2^ (kg/m^2^). Overweight and obesity were defined according to the most recent International Obesity Task Force sex-for-age-BMI cut-offs [36]. Children with a BMI ≥ 95th percentile for age and gender were classified as obese; those with a BMI ≥ 85th but < 95th percentile for age and gender were classified as overweight [36].

### 2.6. Covariates

At recruitment, mothers provided information on age, self-reported height and pre-pregnancy weight, socioeconomic status (proxied by eligibility to the General Medical Services Scheme, a robust indicator of social disadvantage in Ireland) [37], and highest education attainment (tertiary or no tertiary education). Alcohol intake and cigarette smoking during pregnancy were also ascertained using the same questionnaire (current smokers/drinkers and women who have smoked/consumed alcohol in < three months’ time prior to recruitment were classified as exposed). Pre-pregnancy BMI was subsequently derived.

### 2.7. Statistical Analysis

Maternal and paternal characteristics and nutrient intakes were first summarized according to tertiles of HEI-2015 scores and examined using Kruskal–Wallis test for continuous variables or *χ*^2^ test for categorical variables. Relationships between parental HEI scores and nutrient intakes were assessed using Spearman’s correlation. Linear and logistic regression analysis examined associations between parental and grandparental HEI scores with continuous and binary offspring outcomes, respectively. Spearman’s correlation examined associations between grandparental HEI scores and parental HEI scores. The overall trend of odds ratios (ORs) across tertiles of HEI was calculated by considering the median of HEI in each tertile for tertile analysis. Mediation analysis was conducted using the PROCESS macro v2.16 for SPSS (IBM Corporation, Armonk, NY, USA), developed by Hayes, and bias-corrected bootstrap confidence intervals are presented [38]. The regression-based path analysis as a means of estimating various effects of interest (direct and indirect, conditional and unconditional) was implemented using the PROCESS macro available for SPSS [38]. In this study, the mediation model intended to evaluate the direct and indirect effects of GP HEI scores (examined as continuous variables) on offspring outcomes at birth and at 5 years via the intermediary variable of parental HEI scores, with no covariates. Potential confounders and covariates included in our analyses were: Maternal and paternal socioeconomic status, education status, smoking and alcohol intake during pregnancy, energy intake, household income, marital status, physical activity, age at recruitment, pre-pregnancy BMI, child age at follow-up (included in models for 5-year outcomes only), and gender. Missing covariates information was imputed using 20 multiple imputation datasets. All statistical analyses were conducted using SPSS version 24.0 (IBM Corporation, Armonk, NY, USA). Statistical significance was defined as two-sided *p* value < 0.05.

## 3. Results

### 3.1. Characteristics of the Study Population

Table 1 shows the characteristics of Lifeways participants included in this study according to parental tertiles of HEI scores. In 1082 mothers, the mean ± SD maternal HEI score was 52.0 ± 8.6 (range: 19–78). Mothers with higher HEI scores tended to be older, non-smokers, married or cohabiting, and have a higher educational level, household income, and regular physical activity. In 333 fathers, the mean ± SD paternal HEI score was 47.7 ± 9.4 (range: 12–75). Fathers with higher HEI scores tended to be older, and have a higher educational level, household income, and regular physical activity.

### 3.2. Parental HEI Scores and Nutrient Intakes

Table 2 shows significant associations between parental nutrient intakes and HEI scores stratified by tertiles. Both mothers and fathers with higher HEI scores had greater overall caloric intake and higher dietary intake of carbohydrates, protein, PUFA, fibre, calcium, iron, folate, phosphorous, and vitamins C, B6, and E, and lower intake of SFA. Associations between higher HEI scores and higher intake of vitamins B12 and D were observed among the mothers only. Similar results were observed in Spearman’s correlation analysis.

### 3.3. Parental HEI Scores and Offspring Outcomes

Logistic regression analysis examined associations between maternal and paternal HEI tertiles with offspring outcomes at birth and when the child was five years old (Table 3). In the multivariable model, mothers with higher HEI scores were less likely to have delivered infants with LBW (OR 0.72; 95% CI 0.50, 0.99; *p* = 0.04) and had lower likelihood of post-term birth (OR 0.87; 95% CI 0.80, 0.98; *p* = 0.04) in the continuous adjusted model. Among mothers with the highest HEI tertile, risk of both LBW and post-term birth were lower (OR 0.38; 95% CI 0.18, 0.82; *p* = 0.03 and OR 0.50; 95% CI 0.23, 0.98; *p* = 0.03, respectively) compared to mothers with the lowest HEI tertile. However, neither of these findings were statistically significant upon adjustment of covariates. No associations between maternal HEI scores and childhood overweight or obesity at age five were noted. With respect to fathers, in the multivariable adjusted analyses, higher paternal HEI scores were associated with lower odds ratios of childhood obesity at five years old (OR 0.91; 95% CI 0.31, 0.96; *p* = 0.04). This association was even more evident when comparing fathers with highest and lowest HEI tertiles (OR 0.74; 95% CI 0.11, 0.91; *p* = 0.04). It should be noted that the numbers of cases for these outcomes was low and the *p*-trends were not statistically significant for any of the birth outcomes (Table 3). Linear regression analysis revealed significant associations between higher maternal HEI scores and higher birth weight (β 0.98 g; 95% CI 0.01, 1.05 g; *p* = 0.03), longer birth length (β 0.22 cm; 95% CI 0.02, 0.41 cm; *p* = 0.03), and larger head circumference (β 0.30 cm; 95% CI 0.01, 0.51 cm; *p* = 0.04). However, the birth length and head circumference associations were attenuated upon adjustment of covariates (Appendix A).

### 3.4. Grandparental HEI Scores and Offspring Outcomes

Associations between all four maternal and paternal grandparents’ HEI score and that of their grandchild outcomes at birth and age five years were examined. Data from the logistic regression analysis (Table 4) revealed that MGMs with higher HEI scores were less likely to have grandchildren with LBW (OR 0.87; 95% CI 0.61, 0.96; *p* = 0.04) and more likely to have grandchildren with macrosomia (OR 1.10; 95% CI 1.01, 1.22; *p* = 0.03) in the adjusted models, comparing highest to lowest HEI tertiles. The association between MGM HEI scores with lower post-term birth risk (OR 0.51; 95% CI 0.19, 0.98; *p* = 0.03) was attenuated upon adjustment of covariates. PGMs with higher HEI scores were associated with lower risk of overweight grandchildren at five years old (OR 0.84; 95% CI 0.21, 0.98; *p* = 0.04) in the adjusted model. No significant associations were observed between maternal and paternal grandfathers (MGF and PGF) HEI scores with any offspring outcomes at birth or at age five (all *p* > 0.05) (Table 4). Results from the linear regression analysis support these findings. MGMs HEI scores were positively associated with birth weight (β 0.13 g; 95% CI 0.05, 0.30 g; *p* = 0.01) and birth BMI (β 0.08 kg/m^2^; 95% CI 0.04, 0.20 kg/m^2^; *p* = 0.02) after adjusting for potential confounders (Appendix A).

### 3.5. Associations between Parental HEI Scores and Grandparental HEI Scores

Spearman correlation analysis revealed a significant but weak correlation between MGM HEI scores and maternal HEI scores (*r* = 0.21, *p* < 0.01). No significant correlations were observed between maternal and MGF HEI scores or between paternal grandparents HEI and paternal HEI scores.

### 3.6. Mediation Analysis

Figure 1a shows the mediation analysis conducted to assess the degree to which the relationship between MGM HEI scores and grandchild’s birthweight was mediated through maternal HEI scores; a significant direct relationship was observed (β 0.0006, *p* = 0.035). In Figure 1b, mediation analysis with maternal HEI scores as mediator showed a significant direct relationship of MGM HEI scores on low birthweight (β −0.1722, p = 0.045). In Figure 1f, in the paternal line, mediation analysis with paternal HEI scores as mediator showed a significant direct relationship of PGM HEI scores on grandchild’s overweight and obesity status at five years (β −0.0459, *p* = 0.034). No significant associations were observed for parental grandfathers nor for any other offspring outcome.

## 4. Discussion

While maternal nutrition during pregnancy has been previously studied in relation to birth outcomes [1,2], to the best of our knowledge this is the first study to investigate the effect of both parents diet quality on both birth outcomes and weight status, which additionally includes grandparents of both lineages also. In this analysis we show that higher HEI-2015 scores were associated with lower risk of LBW in the maternal and MGM lines, whereas higher paternal and PGM dietary quality predicted lower risk of childhood overweight and obesity at age five. These data suggest that differential intergenerational transmission of risk exists between maternal and paternal lines, whereby maternal line dietary quality appears to influence in utero growth whereas paternal line dietary quality appears to influence postnatal growth.

Poor diet quality during pregnancy is known to increase neonatal adiposity [13,23]. Rodriguez-Bernal et al. found that increasing quintiles of the alternative healthy eating index for pregnancy (AHEI-P) score were associated with higher birth weight in a cohort of Spanish women [16], in line with our findings. However, the results have been inconsistent, with several studies reporting no association between maternal AHEI-P scores, HEI-1995 scores, or adherence to a Mediterranean diet and birthweight or growth [15,18,39]. More recently, a study found that maternal HEI-2010 diet score was not associated with offspring obesity [40]. It should be noted that most studies of maternal diet during pregnancy have used earlier HEI scores [23,24,25], whereas we used the HEI-2015, which may partly contribute to the inconsistent findings reported using the HEI.

These scores indicate a high quality diet with high intake of vegetables, fruits, fibre, protein, and unsaturated fats and low in saturated fats [15,16,17,18,19,20,21,22,23,24]. Higher HEI scores, and thus greater dietary quality, have been strongly associated with a greater dietary variety and intake of fibre, folate, vitamin A and C [23,41]. Also, women with higher HEI scores are more likely to be older, better educated, and have higher incomes, consistent with our findings [16,23,42]. Higher maternal carbohydrate and sugar intakes are associated with higher childhood BMI [9]. Maternal protein intake during pregnancy is not associated with offspring birth weight in an Asian population [43], whereas in another study, higher percentages of energy from protein during pregnancy have been positively associated with birth weight and placental weight [44] and overweight in offspring [8]. In this analysis, we found a positive association between parental HEI and protein intake and furthermore higher maternal HEI was associated with birth weight. Consistent with other analyses of our cohort, we found a significant negative association between maternal SFAs and offspring adiposity [11]. We did not observe an association between maternal diet quality and preterm birth, consistent with results of previous studies [17,23,45]. In contrast, higher dietary quality, determined by the DASH score, was associated with lower risk of preterm birth [19].

The influence of paternal diet quality on offspring outcomes is less studied. We investigated possible relationships between paternal HEI scores and birth outcomes and weight at five years. We identified significant associations between higher paternal HEI scores and lower risk of childhood obesity. Poor paternal diet has been linked with increased risk of metabolic syndrome, obesity, and diabetes in offspring, possibly through epigenetic effects [46,47]. Positive associations have been observed between father–child diet quality using the HEI [26,27] although correlations in mother–child dyads were significantly stronger [27]. Consistent with our findings are the results from a recent study investigating parental nutrient restriction in utero indicating that fetal growth appears to be under matriline influence, but postnatal growth appears to be under patriline intergenerational influences [48].

Our study is highly novel in that it also examined the associations between grandparental HEI scores and grandchild’s outcomes and evaluated the relationships between different generations of HEI scores with birth and childhood outcomes. Mothers and MGMs with higher HEI scores were less likely to have delivered infants with LBW and to have higher birth weight, also consistent with the mediation analysis, and in keeping with an intrauterine or shared environment hypothesis. Notably, there were also associations in the paternal line, with no pattern related to birth outcomes but showing associations between paternal and PGM HEI scores and overweight and obese children at five years of age. These findings persisted in the mediation analysis also. There are no significant associations for grandfathers in either lineage. In a previous analysis of the Lifeways cohort, significant positive correlations were observed for nutrient intakes of maternal grandmother–mother–child triads, but not found in paternal lines [5]. The maternal but not paternal line associations with birth outcomes observed in the current work suggest an intrauterine effect and that confounding by other factors such as environment are less likely, whereas the paternal line associations with childhood weight status may reflect a generally healthier shared environment [49]. A Japanese study showed that MGM and PGM influences on family dietary patterns depended on whether the grandmothers resided or not with the families [50]. Further studies on the different influences of the diet quality across generations in family cohorts are warranted.

Regarding the strengths of the study, this research provides new information about how diet quality is transmitted in different ways across three generations, influencing the health of offspring at birth and childhood. The sample size is reasonably large and we considered many potential confounding factors. The prospective study design is valuable in determining the temporal relationship between exposure (HEI) and offspring outcomes. Further, the HEI-2015 has undergone extensive validation and has been shown to capture the multidimensionality of diet as well as to have predictive validity [22,32]. Nonetheless, a few limitations of our study need to be noted. First, maternal diet in the first trimester was assessed, whereas third-trimester diet may be more relevant for offspring adiposity [51]. However, previous studies indicate that dietary intakes and patterns do not change substantially during pregnancy [52]. Although we controlled for confounding factors, we cannot exclude the possibility that unmeasured confounders may also influence our observations. Moreover, residual confounding arising from imprecise measurement of dietary intake should also be considered. As a structured dietary assessment method, the FFQ is less precise than other methods (i.e., 24-h recalls, food records) and as it is memory based, it can introduce recall and reporting biases. Lastly, the number of fathers with nutrient intake information was relatively low (*n* = 333), potentially leading to lower statistical power and some self-selected bias. Also, there are few families where all four grandparents had information collected. Regarding generalizability of our findings, the Lifeways study was not designed to be representative of the general obstetric population in Ireland [28], though previous analyses suggest that both mothers and grandparents are comparable to the contemporary national health and lifestyles surveys undertaken at the time. In addition, the Food4me study, a multi-centre European study including Irish adults, reported associations between higher HEI-2010 scores with lower BMI and increased physical activity levels [53], consistent with our findings in the parents. Furthermore, a large multi-ethnic cohort study (*n* > 215,000) investigating the predictive validity of HEI-2015 scores confirmed the inverse associations with BMI, risk of mortality, and cardiovascular disease [54].

## 5. Conclusions

In conclusion, we observed that higher HEI-2015 scores were associated with lower risk of LBW in the maternal and MGM lines, whereas higher paternal and PGM dietary quality predicted lower risk of childhood overweight and obesity. These data suggest that differential intergenerational transmission of risk exists between maternal and paternal lineages, meriting further investigation.

## Figures and Tables

**Figure 1 nutrients-11-00928-f001:**
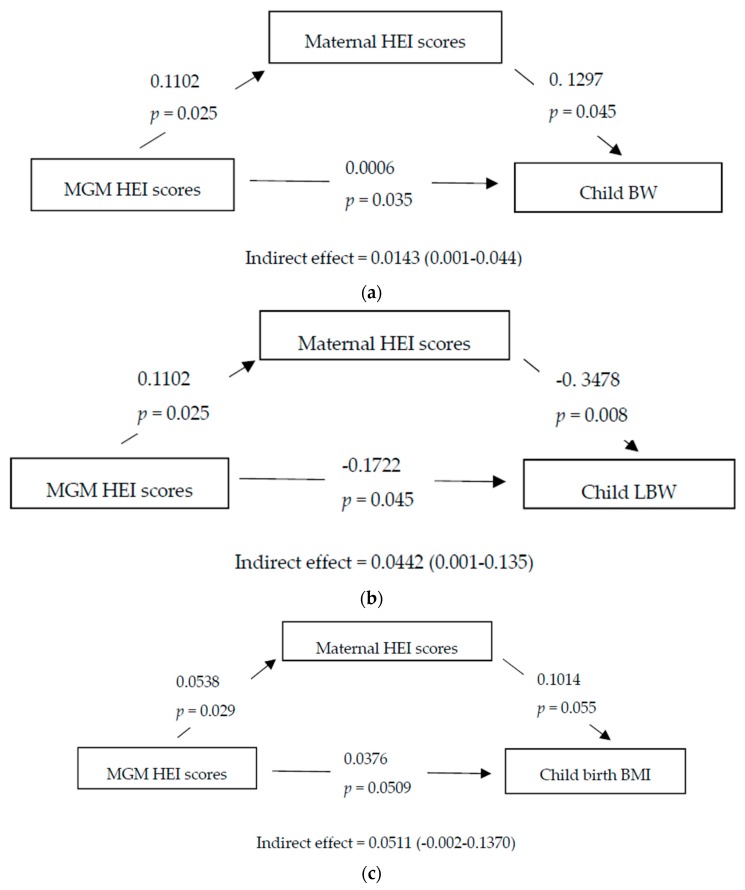
Mediation analysis of grandparental HEI scores on grandchild outcome through parental HEI scores. (**a**) Maternal grandmother (MGM) HEI scores–grandchild birth weight (BW)–maternal HEI scores mediation analysis; (**b**) MGM HEI scores–grandchild low birth weight (LBW)–maternal HEI scores mediation analysis; (**c**) MGM HEI scores–grandchild BMI at five years–maternal HEI scores mediation analysis; (**d**) MGM HEI scores–grandchild macrosomia–maternal HEI scores mediation analysis; (**e**) MGM HEI scores–grandchild post-term birth–maternal HEI scores mediation analysis; (**f**) paternal grandmother (PGM) HEI scores–grandchild overweight and obese (OW/OB) at five years–paternal HEI scores mediation analysis. BW, birthweight; BMI, body mass index; HEI, healthy eating index; LBW, low birth weight; MGM, maternal grandmother; OW/OB, overweight and obese; PGM, paternal grandmother.

**Table 1 nutrients-11-00928-t001:** Characteristics of the study participants according to parental tertiles of healthy eating index (HEI) scores ^1^.

	All Participants	Tertile 1 (Lowest)	Tertile 2	Tertile 3 (Highest)	*p*-Value ^2^
**Maternal Characteristics (*n* = 1082) ^3^**					
Maternal HEI score	52.0 ± 8.6	43.4 ± 4.8	52.8 ± 1.9	62.0 ± 4.1	
Age at mother recruitment (year)	30.1 ± 5.9	28.9 ± 6.0	29.7 ± 6.1	31.9 ± 5.1	**<0.001**
Pre-pregnancy BMI (Kg/m^2^)	23.8 ± 4.2	24.0 ± 4.3	23.9 ± 4.4	23.5 ± 3.7	**0.045**
Height (cm)	163.8 ± 6.4	163.5 ± 6.4	163.3 ± 6.3	164.5 ± 6.4	**0.03**
Education level					
Below tertiary	536 (51)	244 (62)	176 (52)	116 (36)	**<0.001**
Tertiary or above	522 (49)	150 (38)	162 (48)	210 (64)	
Smoking during pregnancy	264 (25)	145 (36)	70 (21)	49 (15)	**<0.001**
Alcohol use during pregnancy	613 (62)	220 (60)	189 (61)	204 (66)	0.29
Marital status					
Married/cohabiting	832 (77)	289 (71)	254 (75)	289 (88)	**<0.001**
Separated/divorced/single	241 (23)	116 (29)	85 (25)	40 (12)	
Household weekly income					
<200£	134 (14)	61 (17)	38 (12)	35 (12)	**0.002**
200–600£	490 (50)	200 (55)	146 (48)	144 (47)	
>600£	348 (36)	101 (28)	121 (40)	126 (41)	
Parity (Non-nulliparous)	585 (55)	221 (54)	185 (55)	179 (55)	0.98
Regular activity	183 (19)	47 (13)	58 (19)	78 (26)	**<0.001**
**Paternal Characteristics (*n* = 333) ^4^**					
Paternal HEI score	47.7 ± 9.4	38.1 ± 4.7	48.0 ± 2.5	58.7 ± 4.8	
Age at proband child birth (year)	33.6 ± 5.5	32.8 ± 5.8	34.0 ± 4.9	34.1 ± 5.7	**0.045**
Paternal BMI (Kg/m^2^)	26.6 ± 3.9	26.3 ± 4.2	27.1 ± 4.3	26.5 ± 3.4	0.53
Height (cm)	178.2 ± 7.1	177.6 ± 7.1	177.6 ± 7.1	179.4 ± 7.0	0.16
Education level					
Below tertiary	160 (48)	72 (60)	54 (50)	34 (33)	**<0.001**
Tertiary or above	172 (52)	48 (40)	55 (50)	69 (67)	
Cigarette smoking	50 (22)	20 (27)	15 (18)	15 (21)	0.38
Alcohol intake	240 (77)	77 (69)	82 (79)	81 (84)	0.05
Marital status					
Married/cohabiting	316 (96)	113 (94)	104 (95)	99 (97)	0.58
Separated/divorced/single	15 (4)	7 (6)	5 (5)	3 (3)	
Household weekly income					
<600£	155 (49)	65 (57)	51 (48)	39 (39)	**0.02**
>600£	164 (51)	48 (43)	55 (52)	61 (61)	
Regular activity	134 (42)	41 (35)	38 (37)	55 (54)	**0.01**

^1^ Values are means ± SD for continuous variables and *n* (%) for categorical variables. ^2^ Kruskal–Wallis test for continuous variables or *χ*^2^ test for categorical variables. ^3^ In mothers missing covariates information: Maternal age (*n* = 52); pre-pregnancy BMI (*n* = 188); height (*n* = 77); education status (*n* = 24); smoking (*n* = 16); alcohol intake (*n* = 98); economic status (*n* = 110); marital status (*n* = 9); parity (*n* = 14); regular activity (*n* = 123). ^4^ In fathers missing covariates information: Paternal age (*n* = 8); paternal BMI (*n* = 37); height (*n* = 17); education status (*n* = 1); smoking (*n* = 107); alcohol intake (*n* = 22); economic status (*n* = 14); marital status (*n* = 2); regular activity (*n* = 10).

**Table 2 nutrients-11-00928-t002:** Parental nutrient intakes according to tertiles of HEI scores ^1^.

	Maternal HEI Scores Tertiles					Paternal HEI Scores Tertiles				
	Tertile 1 (*n* = 411)	Tertile 2 (*n* = 340)	Tertile 3 (*n* = 331)	*p* Value ^2^	Correlation with HEI ^3^	Tertile 1 (*n* = 121)	Tertile 2 (*n* = 109)	Tertile 3 (*n* = 103)	*p* Value ^2^	Correlation with HEI ^3^
Total energy (Kcal/day)	2305.9 ± 1084.6	2511.2 ± 1094.9	2764.2 ± 2119.8	<0.001	0.17 **	2361.9 ± 873.4	2480.6 ± 1082.9	2646.3 ± 837.4	0.02	0.15 **
Carbohydrate (g/day)	273.3 ± 134.2	318.8 ± 121.1	360.0 ± 176.8	<0.001	0.31 **	257.4 ± 103.8	281.6 ± 122.8	321.4 ± 99.3	<0.001	0.28 **
Protein (g/day)	93.9 ± 51.3	105.6 ± 59.2	121.4 ± 143.5	<0.001	0.27 **	97.0 ± 35.5	103.7 ± 34.9	112.0 ± 36.2	0.002	0.20 **
Total fat(g/day)	99.5 ± 52.3	100.9 ± 54.1	102.1 ± 110.6	0.21	−0.06	106.9 ± 44.3	106.1 ± 52.3	102.7 ± 45.9	0.58	−0.06
MUFA (g/day)	32.1 ± 16.9	32.4 ± 17.7	32.6 ± 35.1	0.10	0.07 *	36.3 ± 15.6	36.9 ± 19.7	34.6 ± 15.6	0.36	−0.07
PUFA (g/day)	14.4 ± 9.7	16.2 ± 8.7	18.7 ± 17.1	<0.001	0.20 **	12.9 ± 5.6	15.3 ± 8.6	18.2 ± 13.6	<0.001	0.27 **
SFA (g/day)	40.4 ± 21.6	38.9 ± 21.2	36.2 ± 41.5	<0.001	−0.19 **	44.3 ± 20.1	40.6 ± 20.4	35.0 ± 14.9	0.001	−0.22 **
Cholesterol (mg/day)	314.9 ± 188.3	326.6 ± 193.2	339.9 ± 490.1	0.45	0.01	346.0 ± 159.4	324.2 ± 123.8	310.3 ± 117.5	0.32	−0.10
Fibre (g/day)	21.1 ± 8.7	26.9 ± 9.7	35.4 ± 18.0	<0.001	0.53 **	16.7 ± 6.9	21.1 ± 8.8	29.1 ± 9.9	<0.001	0.57 **
Calcium (mg/day)	969.9 ± 458.7	1077.6 ± 420.1	1222.5 ± 673.4	<0.001	0.21 **	976.3 ± 399.6	1032.7 ± 415.1	1114.5 ± 420.7	0.04	0.13 *
Iron (mg/day)	10.6 ± 5.1	13.6 ± 6.7	17.1 ± 17.5	<0.001	0.45 **	10.3 ± 3.9	12.1 ± 4.4	15.2 ± 5.2	<0.001	0.45 **
Folate (mg/day)	298.0 ± 138.9	368.4 ± 138.5	451.5 ± 248.4	<0.001	0.46 **	267.6 ± 91.6	307.8 ± 100.7	389.4 ± 128.9	<0.001	0.44 **
Phosphorous (mg/day)	1486.9 ± 633.3	1670.7 ± 658.3	1971.5 ± 1569.3	<0.001	0.31 **	1522.3 ± 473.1	1648.7 ± 599.1	1832.0 ± 560.6	<0.001	0.25 **
Vitamin C (mg/day)	125.4 ± 74.0	188.0 ± 103.6	255.9 ± 136.4	<0.001	0.55 **	86.4 ± 51.1	103.8 ± 49.5	163.9 ± 66.5	<0.001	0.53 **
Vitamin B12 (mg/day)	4.8 ± 2.8	5.5 ± 3.9	7.2 ± 8.3	<0.001	0.13 **	5.4 ± 2.9	5.5 ± 2.4	6.0 ± 4.5	0.70	0.06
Vitamin B6 (mg/day)	2.7 ± 1.2	3.3 ± 1.4	3.8 ± 2.8	<0.001	0.38 **	2.7 ± 0.9	3.0 ± 1.1	3.5 ± 1.0	<0.001	0.35 **
Vitamin E (mg/day)	7.9 ± 4.8	9.3 ± 4.1	11.3 ± 7.1	<0.001	0.33 **	7.2 ± 3.3	8.2 ± 3.9	10.8 ± 6.1	<0.001	0.35 **
Vitamin D (mg/day)	3.1 ± 1.9	3.8 ± 2.6	4.2 ± 4.6	<0.001	0.20 **	3.2 ± 1.7	3.5 ± 1.9	3.8 ± 2.3	0.17	0.11

^1^ Values are means ± SD, ^2^ Kruskal–Wallis test, ^3^ Spearman’s correlation coefficients (***p* < 0.01, **p* < 0.05) MUFA (monounsaturated fatty acids), PUFA (polyunsaturated fatty acids), SFA (saturated fatty acids).

**Table 3 nutrients-11-00928-t003:** Associations between parental HEI scores and binary offspring outcomes at birth and at five years ^1^.

	Low Birth Weight		Macrosomia		Preterm Birth		Post-Term Birth		Overweight/Obese at 5 Years		Obese at 5 Years	
*HEI scores*	Unadjusted	Multivariable	Unadjusted	Multivariable	Unadjusted	Multivariable	Unadjusted	Multivariable	Unadjusted	Multivariable	Unadjusted	Multivariable
***Maternal** n(case/total)*												
	52/1070		192/1070		54/958		45/958		191/562		50/562	
T2	0.62 (0.32, 1.18)	0.82 (0.37, 1.82)	1.16 (0.79, 1.70)	1.15 (0.73, 1.76)	0.81 (0.42, 1.57)	0.89 (0.41, 1.87)	0.60 (0.29, 1.23)	0.84 (0.45, 1.81)	0.89 (0.58, 1.36)	1.21 (0.71, 2.04)	1.37 (0.65, 2.89)	1.10 (0.40, 3.42)
T3	**0.38** **(0.18, 0.82) ***	0.53 (0.19, 1.02)	1.30 (0.89, 1.90)	1.21 (0.54, 1.74)	0.76 (0.39, 1.49)	0.67 (0.31, 1.64)	**0.50** **(0.23, 0.98) ***	0.73 (0.25, 1.12)	0.87 (0.57, 1.33)	0.97 (0.56, 1.62)	1.58 (0.77, 3.23)	1.12 (0.65, 2.99)
Cont.	**0.61** **(0.43, 0.90) ***	**0.72** **(0.50, 0.99) ***	1.01 (0.99, 1.03)	1.02 (0.89, 1.14)	0.98 (0.94, 1.01)	0.63 (0.51, 1.10)	**0.96** **(0.93, 0.99) ***	**0.87** **(0.80, 0.98) ***	0.99 (0.97, 1.02)	1.02 (0.99, 1.08)	1.02 (0.98, 1.05)	1.01 (0.94, 1.10)
*P* _trend_	**0.01**	**0.04**	0.36	0.45	0.15	0.23	**0.03**	**0.04**	0.83	0.66	0.28	0.31
***Paternal** n(case/total)*												
	9/329		67/329		14/305		17/305		71/218		17/218	
T2	1.54 (0.45, 2.51)	1.64 (0.24, 11.36)	1.23 (0.62, 2.47)	1.34 (0.53, 3.35)	**1.73** **(1.31, 2.31) ***	1.29 (0.20, 1.94)	1.11 (0.59, 2.08)	1.42 (0.93, 2.18)	1.41 (0.67, 2.82)	1.51 (0.66, 3.40)	1.16 (0.84, 2.14)	1.00 (0.51, 1.39)
T3	0.66 (0.38, 1.14))	0.65 (0.10, 8.14)	1.41 (0.54, 2.54)	1.52 (0.58, 3.96)	1.24 (0.91, 1.68)	1.61 (0.10, 2.42)	0.79 (0.40, 1.56)	0.76 (0.47, 1.22)	1.22 (0.26, 3.05)	1.42 (0.60, 3.42)	0.89 (0.26, 2.77)	**0.74** **(0.11, 0.91) ***
Cont.	0.92 (0.77, 1.11)	0.64 (0.33, 1.21)	1.18 (0.85, 1.65)	0.98 (0.70, 1.37)	1.11 (0.96, 1.28)	1.24 (0.98, 1.52)	1.11 (0.61, 2.03)	0.61 (0.48, 1.78)	1.05 (0.74, 1.50)	0.93 (0.61, 1.10)	1.03 (0.42, 1.26)	**0.91** **(0.31, 0.96) ***
*P* _trend_	0.92	0.49	0.31	0.47	0.14	0.08	0.72	0.50	0.76	0.10	0.14	**0.04**

^1^ Values are OR (95% CI) expressed for 10-point increment in HEI scores for continuous analysis and with reference to the lowest tertile (T1) for tertile analysis. Multivariable models were adjusted for maternal and paternal socio-economic status, education status, marital status, cigarettes smoking and alcohol consumption, energy intake, household income, age at recruitment, pre-pregnancy body mass index, and child age at follow-up (included in models for five-year outcomes only) and sex (for overweight and obese status child sex was intrinsically adjusted). Missing covariate information was handled by pooling effect estimates from 20 multiply-imputed datasets. **p* < 0.05.

**Table 4 nutrients-11-00928-t004:** Associations between grandparental HEI scores and binary offspring outcomes at birth and at five years ^1^.

	Low Birth Weight		Macrosomia		Preterm Birth		Post-Term Birth		Overweight/Obese at 5 years		Obese at 5 years	
*HEI scores*	Unadjusted	Multivariable	Unadjusted	Multivariable	Unadjusted	Multivariable	Unadjusted	Multivariable	Unadjusted	Multivariable	Unadjusted	Multivariable
***MGM*** *n (case/total)*	10/281		53/281		15/259		13/259		52/177		12/177	
T2	0.92 (0.12, 1.17)	0.87 (0.12, 1.64)	1.18 (0.58, 1.81)	1.13 (0.73, 1.67)	0.86 (0.40, 1.84)	0.90 (0.41, 1.45)	0.32 (0.14, 2.54)	0.25 (0.01, 1.81)	0.65 (0.24, 1.19)	0.68 (0.29, 1.30)	0.89 (0.17, 1.85)	0.90 (0.17, 1.59)
T3	**0.72** **(0.23, 0.98) ***	0.95 (0.12, 1.05)	1.01 (0.70, 1.43)	0.64 (0.59, 1.16)	0.59 (0.04, 1.25)	0.52 (0.21, 1.27)	0.47 (0.27, 1.22)	0.87 (0.56, 1.25)	0.56 (0.22, 1.21)	0.58 (0.24, 1.24)	0.81 (0.25, 2.03)	0.85 (0.27, 2.07)
Cont.	0.97 (0.57, 1.02)	**0.87** **(0.61, 0.96) ***	0.97 (0.10, 1.27)	**1.10** **(1.01, 1.22) ***	0.75 (0.54, 1.24)	0.74 (0.53, 1.05)	**0.51** **(0.19, 0.98) ***	0.67 (0.14, 1.53)	0.79 (0.61, 1.20)	0.82 (0.58, 1.37)	0.90 (0.60, 1.32)	0.95 (0.54, 1.40)
*P* _trend_	0.08	**0.04**	0.54	**0.03**	0.36	0.17	**0.03**	0.73	0.14	0.17	0.31	0.28
***MGF*** *n (case/total)*	5/112		20/112		4/105		3/105		23/71		2/71	
T2	0.94 (0.22, 1.92)	0.87 (0.15, 1.68)	0.98 (0.38, 1.16)	0.85 (0.48, 1.26)	0.20 (0.10, 1.12)	0.25 (0.09, 1.31)	0.84 (0.29, 2.18)	0.94 (0.10, 2.07)	1.04 (0.39, 2.74)	1.10 (0.28, 2.51)	1.11 (0.31, 2.17)	1.10 (0.37, 2.69)
T3	1.10 (0.38, 2.03)	0.74 (0.25, 1.88)	1.07 (0.58, 2.04)	0.98 (0.34, 1.46)	0.57 (0.16, 1.28)	0.46 (0.10, 1.08)	0.79 (0.20, 1.56)	0.77 (0.23, 1.98)	0.66 (0.13, 2.33)	0.77 (0.21, 2.09)	0.89 (0.26, 2.47)	0.88 (0.21, 2.15)
Cont.	1.21 (0.70, 2.44)	0.87 (0.30, 2.13)	1.03 (0.77, 2.27)	0.99 (0.65, 2.31)	0.78 (0.21, 1.22)	0.65 (0.26, 1.26)	0.36 (0.14, 1.26)	0.41 (0.59, 1.88)	0.82 (0.13, 1.48)	0.99 (0.18, 1.234)	0.95 (0.42, 1.65)	0.96 (0.55, 1.79)
*P* _trend_	0.83	0.68	0.25	0.41	0.30	0.46	0.34	0.39	0.19	0.29	0.88	0.58
***PGM*** *n(case/total)*	4/161		25/161		4/147		5/147		30/97		7/97	
T2	1.14 (0.23, 2.75)	1.07 (0.14, 2.71)	1.01 (0.47, 1.91)	0.99 (0.36, 2.38)	1.23 (0.64, 2.25)	1.02 (0.38, 2.55)	0.62 (0.17, 1.85)	0.90 (0.42, 1.87)	0.48 (0.11, 1.60)	0.46 (0.15, 1.89)	1.03 (0.54, 2.62)	0.99 (0.60, 1.82)
T3	0.81 (0.33, 1.99)	0.72 (0.20, 1.90)	1.07 (0.51, 2.08)	1.10 (0.54, 1.91)	1.15 (0.42, 1.85)	1.13 (0.57, 2.49)	0.43 (0.27, 1.21)	0.51 (0.22, 1.38)	0.65 (0.15, 1.14)	0.67 (0.32, 1.08)	0.88 (0.35, 1.76)	0.79 (0.58, 1.49)
Cont.	0.93 (0.27, 2.01)	0.92 (0.23, 1.99)	1.24 (0.40, 2.38)	1.16 (0.48, 2.27)	1.06 (0.58, 1.95)	1.18 (0.80, 2.53)	0.89 (0.38, 1.53)	0.75 (0.54, 1.17)	**0.74** **(0.14, 0.92) ***	**0.84** **(0.21, 0.98) ***	0.97 (0.41, 1.32)	1.00 (0.51, 1.47)
*P* _trend_	0.11	0.15	0.68	0.51	0.24	0.33	0.63	0.24	**0.02**	**0.04**	0.28	0.42
***PGF*** *n(case/total)*	4/98		11/98		5/86		3/86		19/58		2/58	
T2	0.77 (0.24, 1.67)	0.67 (0.19, 1.54)	1.14 (0.37, 1.75)	0.95 (0.17, 1.87)	0.56 (0.14, 1.55)	0.51 (0.25, 1.57)	0.67 (0.31, 1.74)	0.60 (0.28, 1.82)	0.79 (0.34, 2.20)	0.87 (0.52, 2.11)	0.89 (0.54, 2.11)	0.79 (0.54, 2.25)
T3	0.84 (0.34, 1.86)	0.75 (0.15, 1.97)	1.08 (0.55, 1.96)	1.07 (0.64, 1.82)	0.71 (0.21, 1.81)	0.62 (0.23, 1.63)	0.75 (0.15, 2.31)	0.71 (0.13, 1.79)	0.98 (0.46, 2.18)	0.90 (0.35, 2.06)	0.97 (0.61, 1.97)	0.82 (0.51, 2.07)
Cont.	0.93 (0.43, 1.64)	1.10 (0.67, 2.18)	0.71 (0.47, 1.97)	0.78 (0.56 1.89)	0.76 (0.45, 1.89)	0.67 (0.19, 2.12)	0.73 (0.25, 2.30)	0.78 (0.43, 1.95)	1.23 (0.41, 2.14)	0.84 (0.22, 2.21)	1.01 (0.64, 2.86)	0.96 (0.31, 2.08)
*P* _trend_	0.32	0.27	0.49	0.31	0.41	0.19	0.19	0.21	0.48	0.61	0.79	0.64

^1^ Values are OR (95% CI) expressed for 10-point increment in HEI scores for continuous analysis and with reference to the lowest tertile (T1) for tertile analysis. Multivariable models were adjusted for socio-economic status, education status, marital status, cigarettes smoking and alcohol consumption, age at recruitment, energy intake, household income, body mass index, and child age at follow-up (included in models for five-year outcomes only) and sex (for overweight and obese status child sex was intrinsically adjusted). Missing covariate information was handled by pooling effect estimates from 20 multiply-imputed datasets. **p* < 0.05.

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
