# Peer review of "Adherence to the Healthy Eating Index-2015 across Generations Is Associated with Birth Outcomes and Weight Status at Age 5 in the Lifeways Cross-Generation Cohort Study"

_nutrients, 2019, doi:10.3390/nu11040928_

Round 1

Reviewer 1 Report

My overall impression of the paper is that a lot of work has been put into using the wealth of data collected for one paper versus having a more focused paper and multiple papers. For example, literature to date is fairly clear that children in this age group eat what their mothers eat. It would be interesting to look at father eating and compare to mothers eating (for one paper). Less is known re the concordance of mother and fathers HEI scores. I feel the piece on grandparents diet is a bit of a stretch b/c I am unsure how grandparents diet is directly related to the child’s weight status besides serving a role as a mediator (which the authors have included). I think this is an interesting paper (the mediation analysis), but on its own. I also think that how HEI from the mom’s diet is related to offspring weight is relevant and acceptable as own paper. I feel if the father’s and grandparents diet are concordant with the mothers diet, then that is one thing, but I do not think you can take it further and say that there is an association btw grandparents HEI and offspring outcomes (hence Table 4 is not an appropriate analysis in my opinion). Grandparents diet may be correlated with mom’s diet and then mom’s diet is correlated with her weight gain/health during pregnancy which results in specific offspring weight-related outcomes at birth.

With this said, I think the authors have an incredible data set and the ability to address multiple questions to move forward research in nutrition epidemiology. I suggest that the authors re-focus the paper using some of my ideas above or their own ideas and then produce a more succinct manuscript.

Author Response

Thank you for reviewing our manuscript. We are glad the reviewer acknowledges the extensive analyses performed and appreciates the value of the Lifeways dataset in the context of investigating potential intergenerational influences of diet on offspring outcomes. Three generation birth cohorts are relatively rare. Lifeways is one of very few cohorts worldwide to have detailed dietary, lifestyle, clinical, anthropometric and sociodemographic data from parents and grandparents, collected directly and prospectively. Examination of the whole cohort allows us to advance the state of the art beyond maternal diet-offspring birth outcome associations by additionally examining fathers as well as grandparents of both lineages. We further expand on this by investigating postnatal growth. Our analysis maximises the resource available to us and allows us to comprehensively examine whether differential transmission of risk exists between maternal and paternal lines in the context of offspring fetal and postnatal growth. Our analysis is focused on addressing our objective, as stated in the introduction “…to investigate any associations between maternal, paternal and grandparental dietary quality, determined by HEI-2015 scores, and offspring birth outcomes and weight status at 5 years...”

Thanks for the suggestion regarding correlations between parental-grandparental HEI scores. We have conducted that analysis and observed a significant but weak positive correlation between MGM HEI scores and maternal HEI scores. We have included these results in lines 221-224:

3.5. Associations between parental HEI scores and grandparental HEI scores

Spearman correlation analyses shows a significant but weak correlation between maternal and MGM HEI scores (r=0.21, p<0.01). No significant correlations were observed with MGF HEI scores or between paternal and paternal grandparents HEI scores.

Reviewer 2 Report

This is a well-written report on the association between parental and grandparental diet quality and child birth outcomes and weight at 5 years. I believe this report would be of high interest to the journal’s audience; however, some clarifications are needed in the methods, and some additional analyses need to be performed to make the results coherent and complete. Following are my specific suggestions for the authors.

1.       Line 113: What is the range of the actual ages at the follow-up visit? If the range is wide (e.g. 4.5 to 5.5 years old), an adjustment for the actual age at the follow-up visit may be required.

2.       Line 117: please state those 85th and 95th percentiles used as cut-offs.

3.       Lines 134-135: it is unclear what the authors mean by ‘with reference to the lowest tertile’ for continuous variables. Please clarify or delete.

4.       Lines 135-136: rather than just referring to the macro, please state the methods implemented in the macro (and used in the present analysis).

5.       Lines 137-140: were all listed covariates included in all models? If not, how were covariates selected for inclusion?

6.       Table 3: the total sample size for each outcome vs. maternal HEI is given in the column headings. However, the total sample size for these outcomes vs. paternal HEI is not stated. Since the number of fathers enrolled (n=333) was much smaller than the number of mothers, these paternal HEI models conceivably have smaller sample size, since they include only a subset of enrolled children whose fathers are also in the study (correct?). Please add the sample sizes for paternal models to Table 3.

7.       Table 3: the authors state in the text that the number of observed events for some adverse birth outcomes were small, however they do not state these numbers. Please add the total number of events observed for each of the outcomes in Table 3, separately for maternal and paternal models.

8.       Lines 184-189: linear regression results for birth weight, length and head circumference are not included as a table but would be of great interest to the reader. Please include these (both unadjusted and multivariable models) as a separate table, possibly as a supplementary table.

9.       Table 4: as with Table 3, please add the total number of observed events for each outcome/each model type (MGM, MGF, PGM, PGF).

10.   Table 4: all grandparent HEI models are not adjusted for the corresponding parent’s HEI (e.g. MGM, MGF for mother’s HEI and PGM, PGF for father’s HEI), which makes one question whether the observed associations are really due to grandparent’s HEI or due to its correlation with a parent’s HEI. These models need to be re-run with adjustment for the parent’s HEI, and the results reported. The mediation analysis was only conducted for BW/MGM and OW/OB at 5 y/PGM, not for other outcomes that showed statistically significant associations (MGM vs. low BW, macrosomia, post-term birth), and thus does not sufficiently address this concern.

11.   Lines 206-209: As with maternal and paternal linear regression models, the linear regression models relating grandparental HEI to birth anthropometrics needs to be presented as a separate, perhaps supplementary table.

12.   Section 3.5: more details are needed in this section. Was mediation analysis conducted using HEI as a continuous variable?

13.   In the PGM vs. child OW/OB at 5 years, were the OW and OB outcomes pooled together? If so, this presents a disjoint with Table 4, where these outcomes are separate. In other words, mediation analysis is conducted for a model other than those presented in Table 3. This needs to be rectified, with mediation analysis for the same models as in Table 3.

14.   Mediation analysis for other outcomes with statistically significant associations (see comment #10 above) needs to be conducted as well and the results presented.

15.   Maternal and paternal HEI are examined in separate models, however it is conceivable that they might have synergistic effect on the birth and 5-year outcomes through shared environment (as the authors hypothesize regarding PGM on lines 306-308). The authors need to examine their joint effect by fitting models that include birth maternal and paternal HEI and their interaction.

Minor editorial comments:

16.   Lines 16-18: Please edit the sentence for grammar.

17.   Abstract: please mention study location.

18.   ‘Fetal’ (line 28) vs. ‘foetal’ (line 35): please choose one (suggest ‘fetal’ as the scientifically preferred version) and be consistent throughout.

19.   Line 52: please replace with “adherence to”.

Author Response

Reviewer 2

Comments and Suggestions for Authors

This is a well-written report on the association between parental and grandparental diet quality and child birth outcomes and weight at 5 years. I believe this report would be of high interest to the journal’s audience; however, some clarifications are needed in the methods, and some additional analyses need to be performed to make the results coherent and complete. Following are my specific suggestions for the authors.

Response:

Thank you for your critical appraisal of our manuscript and constructive comments. We are glad that you found our manuscript to be well written and of high interest to Nutrients readers. We have addressed each of the comments to the best of our ability and revised the manuscript accordingly. We feel the manuscript has been strengthened as a result of these modifications and clarifications.

1. Line 113: What is the range of the actual ages at the follow-up visit? If the range is wide (e.g. 4.5 to 5.5 years old), an adjustment for the actual age at the follow-up visit may be required.

Response:

The children age range is 4.6 to 6.2, mean: 5.4 years old. We have now included child age in the 5 year analysis and report that the results did not materially change. We have edited the relevant results and footnotes in the tables (Table 3, 4, S1 and S2) and text in the methods section.  

2. Line 117: please state those 85th and 95th percentiles used as cut-offs.

Response:

Thanks for the suggestion and we have modified the sentence in line 116-119:

 “Overweight and obesity were defined according to the most recent International Obesity Task Force sex-for-age-BMI cut-offs [37]. Children with a BMI ≥95th percentile for age and gender were classified as obese; those with a BMI ≥85th but < 95th percentile for age and gender were classified as overweight [37].”

3. Lines 134-135: it is unclear what the authors mean by ‘with reference to the lowest tertile’ for continuous variables. Please clarify or delete.

Response:

Thanks for the opportunity to clarify. We have modified the sentence in line 136-138:

“The overall trend of odds ratios (ORs) across tertiles of HEI was calculated by considering the median of HEI in each tertile with reference to the lowest tertile (T1) for tertile analysis.”

4. Lines 135-136: rather than just referring to the macro, please state the methods implemented in the macro (and used in the present analysis).

Response:

We have included the methods implemented in the PROCESS macro in lines 138-142:

“Mediation analysis was conducted using the PROCESS macro v2.16 for SPSS, developed by Andrew Hayes and bias-corrected bootstrap confidence intervals are presented [39]. In this study, the mediation model intended to evaluate the direct and indirect effects of GP HEI scores (examined as continuous variables) on offspring outcomes at birth and at 5 years via the intermediary variable of parental HEI scores.”

5. Lines 137-140: were all listed covariates included in all models? If not, how were covariates selected for inclusion?

Response:

Yes, all listed covariates were included in all models.

6. Table 3: the total sample size for each outcome vs. maternal HEI is given in the column headings. However, the total sample size for these outcomes vs. paternal HEI is not stated. Since the number of fathers enrolled (n=333) was much smaller than the number of mothers, these paternal HEI models conceivably have smaller sample size, since they include only a subset of enrolled children whose fathers are also in the study (correct?). Please add the sample sizes for paternal models to Table 3.

Response:

Thanks for the suggestion. We have included the sample size for paternal models in Table 3 as requested.

7. Table 3: the authors state in the text that the number of observed events for some adverse birth outcomes were small, however they do not state these numbers. Please add the total number of events observed for each of the outcomes in Table 3, separately for maternal and paternal models.

Response:

Thanks for the suggestion. We have included the number of observed events for adverse birth outcomes (case/total) for maternal and paternal models in Table 3 as requested.

8. Lines 184-189: linear regression results for birth weight, length and head circumference are not included as a table but would be of great interest to the reader. Please include these (both unadjusted and multivariable models) as a separate table, possibly as a supplementary table.

Response:

Thanks for the suggestion. We have included the linear regression results for parental HEI scores and offspring outcomes in Supplemental Table S1.

9. Table 4: as with Table 3, please add the total number of observed events for each outcome/each model type (MGM, MGF, PGM, PGF).

Response:

We have included the numbers of observed events for adverse birth outcomes for the grandparental models in Table 4 as requested.

10. Table 4: all grandparent HEI models are not adjusted for the corresponding parent’s HEI (e.g. MGM, MGF for mother’s HEI and PGM, PGF for father’s HEI), which makes one question whether the observed associations are really due to grandparent’s HEI or due to its correlation with a parent’s HEI. These models need to be re-run with adjustment for the parent’s HEI, and the results reported. The mediation analysis was only conducted for BW/MGM and OW/OB at 5 y/PGM, not for other outcomes that showed statistically significant associations (MGM vs. low BW, macrosomia, post-term birth), and thus does not sufficiently address this concern.

Response:

Thanks for the suggestion. We have now assessed the correlations between grandparental HEI scores and parental HEI scores. We observed a significant weak positive correlation between MGM HEI scores and maternal HEI scores. No significant correlations were observed with the MGF HEI scores or between paternal grandparents HEI and paternal HEI scores. The models were re-run with adjustment for the parent´s HEI scores and the results did not materially change. We have included this additional information on page 13 lines 221-224.

In relation to the second part of the comment, mediation analysis has now been conducted for other offspring outcomes (MGM vs. low BW, macrosomia, post-term birth, and MGM vs. child birth BMI as requested) and these models have been included in Figure 1.

11. Lines 206-209: As with maternal and paternal linear regression models, the linear regression models relating grandparental HEI to birth anthropometrics needs to be presented as a separate, perhaps supplementary table.

 Response:

We have included linear regression results for grandparental HEI scores and offspring outcomes as requested in Supplemental Table S2.

12.   Section 3.5: more details are needed in this section. Was mediation analysis conducted using HEI as a continuous variable?

Response:

Thanks for the opportunity to clarify. We have addressed this comment in our response to comment 4. “In this study, the mediation model intended to evaluate the direct and indirect effects of GP HEI scores (examined as continuous variables) on offspring outcomes at birth and at 5 years via the intermediary variable of parental HEI scores.” (Included in methods on page 4 lines 140-141).

13. In the PGM vs. child OW/OB at 5 years, were the OW and OB outcomes pooled together? If so, this presents a disjoint with Table 4, where these outcomes are separate. In other words, mediation analysis is conducted for a model other than those presented in Table 3. This needs to be rectified, with mediation analysis for the same models as in Table 3.

Response:

Apologies for this oversight with the column sub-headings and any confusion arising. All analyses were conducted with combined overweight/obese and obese as outcomes. These subheadings have been corrected in both Table 3 and 4 to read as Overweight/Obese and Obese.

14. Mediation analysis for other outcomes with statistically significant associations (see comment #10 above) needs to be conducted as well and the results presented.

Response:

Mediation analysis was conducted for other offspring outcomes (MGM vs. low BW, macrosomia, post-term birth, and MGM vs. child birth BMI as suggested). These data have been included in Figure 1.

“…Figure 1b, mediation analysis with maternal HEI scores as mediator showed a significant direct relationship of MGM HEI scores on low birthweight (β -0.1722, P=0.045). In Figure 1f, in the paternal line, mediation analysis with paternal HEI scores as mediator showed a significant direct relationship of PGM HEI scores on grandchild’s overweight and obesity status at 5 years (β -0.0459, P=0.034). No significant associations were observed for parental grandfathers nor for any other offspring outcome.” (lines 228-233)

15. Maternal and paternal HEI are examined in separate models, however it is conceivable that they might have synergistic effect on the birth and 5-year outcomes through shared environment (as the authors hypothesize regarding PGM on lines 306-308). The authors need to examine their joint effect by fitting models that include birth maternal and paternal HEI and their interaction.

Response:

We aimed to compare maternal and paternal associations with offspring outcomes, with a view to determining whether differential transmission of risk exists between maternal and paternal lines in the context of offspring fetal and postnatal growth, to this end we examined both parent HEI-offspring outcomes separately. Had we observed similar associations for both mothers and fathers with offspring outcomes that might indicate a shared environmental or genetic effect, however we observed differential associations between maternal and paternal lines.The maternal but not paternal line associations observed with birth outcomes suggest an intrauterine effect and that confounding by other factors such as environment is less likely, whereas the paternal line associations with childhood weight status may reflect a generally healthier shared environment (ref: Lawlor, D.A.; Leary, S.; Smith, G.D. Theoretical underpinning for the use of intergenerational studies in life course epidemiology. In: Lawlor DA, Mishra GD (eds). Family matters: designing, analysing and understanding family based studies in life course epidemiology. Oxford University Press, UK, 2009, pp. 13–38.).

We have edited the text (on page 15 lines 351-354) to strengthen this point:

The maternal but not paternal line associations with birth outcomes observed in the current work suggest an intrauterine effect and that confounding by other factors such as environment is less likely, whereas the paternal line associations with childhood weight status may reflect a generally healthier shared environment [50].

Minor editorial comments:

16. Lines 16-18: Please edit the sentence for grammar.

Response:

Thanks for the suggestion and we have edited the sentence in line 16-18:

“The Lifeways Cohort Study in the Republic of Ireland comprises 1082 index-child’s mothers, 333 index-child’s fathers and 707 grandparents.”

17. Abstract: please mention study location.

Response:

The study location is now mentioned in the abstract in line 17:

 “The Lifeways Cohort Study in the Republic of Ireland…”

18. ‘Fetal’ (line 28) vs. ‘foetal’ (line 35): please choose one (suggest ‘fetal’ as the scientifically preferred version) and be consistent throughout.

Response:

Thank you for the suggestion. We have now changed “foetal” to “fetal”.

19. Line 52: please replace with “adherence to”.

Response:

Thanks for the opportunity to clarify and we have modified the sentence according to your suggestion.

Reviewer 3 Report

This paper describes the transgenerational effects of diet quality on child outcomes. The authors describe a number of interesting findings that indicate differential intergenerational transmission via maternal and paternal lines.

In general, I found this paper to be sound in rationale, design, presentation of results, and conclusions. My major question is why the authors chose to make the biological sex of the child a co-variate. In non-human species, there is plenty of evidence to suggest maternal diet has sex-specific effects on offspring. It seems unfortunate not to take advantage of the ability to include the sex of the child as a factor in analyses. What was the basis for this decision?

Typos, etc:

Very few, the paper was well-written.

Abstract - lines 23, 24 - should read maternal and paternal

Author Response

Reviewer 3

Comments and Suggestions for Authors

This paper describes the transgenerational effects of diet quality on child outcomes. The authors describe a number of interesting findings that indicate differential intergenerational transmission via maternal and paternal lines.

In general, I found this paper to be sound in rationale, design, presentation of results, and conclusions. My major question is why the authors chose to make the biological sex of the child a co-variate. In non-human species, there is plenty of evidence to suggest maternal diet has sex-specific effects on offspring. It seems unfortunate not to take advantage of the ability to include the sex of the child as a factor in analyses. What was the basis for this decision?

Response:

Thank you for reviewing our manuscript. We are glad that you found our results of interest and our paper to be well designed and presented.

Based on previous reports from similar studies about maternal diet quality and fetal growth (references 13,16, 23) as well as previous investigation of maternal diet and offspring weight status in our Lifeways cohort (reference 11) we chose to include offspring gender in our analyses as a covariate only. Given the variation in numbers of participating mothers, fathers and grandparents, performing gender stratified analysis may negatively impact on statistical power to detect meaningful associations where limited numbers of certain family members have participated. We acknowledge the important point made by the reviewer and examined, for the reviewers information only, whether any gender differences with either the maternal-birth and paternal-childhood outcome associations exist. No gender specific associations were identified. Going forward we will address this issue in future investigations (for example in large-scale pooled analysis of birth cohorts) which will be sufficiently statistically powered to detect gender specific associations.  

Abstract - lines 23, 24 - should read maternal and paternal

It's correct....paternal HEI scores and paternal grandmothers (PGM) HEI scores

Round 2

Reviewer 1 Report

None. 

Author Response

We would like to thanks the reviewer 1.

Reviewer 2 Report

The authors have adequately addressed most of my prior comments and have clarified their methods and results. However, a few of my prior comments still require attention.

1.      Original comment #3, line 151: The revised sentence is still unclear. Please delete the part starting with “with reference to the …”.

2.      Original comment #4, lines 152-153: the revised sentence does not describe the method implemented by the macro, as originally requested. It states the type of confidence intervals used and the goal of the mediation analysis (addressing my comment #12), but does not state the method implemented by the macro. Please state this method.

3.      Original comment #13, Table 4: The authors have replaced the column heading for the Overweight at 5 y with “Overweight/Obese at 5y”, which I assume (and hope) was what they originally had modeled. However, as it now stands, for the PGF model the number of events for OW/Obese is 26, while the number of events for Obese only is 27, which is impossible and makes the entire table doubtful. Please correct this. I hope this is only a typo.

Additional comments:

4.      The authors claim to have additionally adjusted all multivariable models for the actual child age at 5-year follow-up. However, all results in Table 3 look exactly identical, with no change to any estimate or CI even in the 2nd decimal point. This is really hard to believe. Further, in Table 4, all results are identical to the previous version, except for post-term birth models, both unadjusted and adjusted. This brings up several questions. First, why are the unadjusted results for post-term birth in Table 4 different? There was no additional adjustment in the unadjusted models. Second, why was the post-term model adjusted for child age at 5-year follow-up, which is irrelevant for this model? And third, considering all this,how is it possible that the results for OW/Obese and Obese at 5 years did not change at all after additional adjustment? Was this additional adjustment actually made for the 5-year models? Only these models had to be adjusted for child age at 5-year follow-up. Please double check and adjust if it was somehow not done due to errors in programming, etc. This refers to both Tables 3 and 4.

Author Response

The authors have adequately addressed most of my prior comments and have clarified their methods and results. However, a few of my prior comments still require attention.

1.      Original comment #3, line 151: The revised sentence is still unclear. Please delete the part starting with “with reference to the …”.

Response 1: Have deleted “with reference to the …”.as suggested.

2.      Original comment #4, lines 152-153: the revised sentence does not describe the method implemented by the macro, as originally requested. It states the type of confidence intervals used and the goal of the mediation analysis (addressing my comment #12), but does not state the method implemented by the macro. Please state this method.

Response 2: Thanks for the opportunity to clarify. We have included now the methods implemented in the PROCESS macro in lines 139-144.

“…The regression-based path analysis as a means of estimating various effects of interest (direct and indirect, conditional and unconditional) was implemented using the PROCESS macro available for SPSS [39]. In this study, the mediation model intended to evaluate the direct and indirect effects of GP HEI scores (examined as continuous variables) on offspring outcomes at birth and at 5 years via the intermediary variable of parental HEI scores, with no covariates.”

3.      Original comment #13, Table 4: The authors have replaced the column heading for the Overweight at 5 y with “Overweight/Obese at 5y”, which I assume (and hope) was what they originally had modeled. However, as it now stands, for the PGF model the number of events for OW/Obese is 26, while the number of events for Obese only is 27, which is impossible and makes the entire table doubtful. Please correct this. I hope this is only a typo.

Response 3: Thanks for the opportunity to clarify and apologies for this oversight. The reviewer is correct in thinking that our original model was indeed for the combined overweight and obese at 5yr. We have checked Table 4 for accuracy. The number of events reported for GP HEI scores and Overweight/Obese and Obese is incorrect. This was a typo. We have now corrected the numbers of events in Overweight/Obese and Obese at 5y in Table 4.

Additional comments:

4.      The authors claim to have additionally adjusted all multivariable models for the actual child age at 5-year follow-up. However, all results in Table 3 look exactly identical, with no change to any estimate or CI even in the 2nd decimal point. This is really hard to believe. Further, in Table 4, all results are identical to the previous version, except for post-term birth models, both unadjusted and adjusted. This brings up several questions. First, why are the unadjusted results for post-term birth in Table 4 different? There was no additional adjustment in the unadjusted models. Second, why was the post-term model adjusted for child age at 5-year follow-up, which is irrelevant for this model? And third, considering all this, how is it possible that the results for OW/Obese and Obese at 5 years did not change at all after additional adjustment? Was this additional adjustment actually made for the 5-year models? Only these models had to be adjusted for child age at 5-year follow-up. Please double check and adjust if it was somehow not done due to errors in programming, etc. This refers to both Tables 3 and 4.

Response 4: We appreciate the exhaustive review and the suggested comments. Please let us clarify. We re-ran our analysis of the adjusted 5 year models additionally adjusting for the child age at 5-year follow-up as suggested in the first review. We have clarified in the methods (page 4 lines 147) and also in the footnotes of Table 3 and 4 that child age at follow-up was only included in the 5 year models. As reported in our previous response the findings did not materially change with this additional adjustment. We double checked the content of Table 3 and 4 in our last submission. Unfortunately, these tables did not contain the new results for the adjusted 5 year models. We also found typographical errors and duplicated results for the post-term birth results. We apologise for these errors and thank the reviewer for highlighting them. We have now corrected the results in both Tables 3 and 4 and have ensured that the correct and fully finalised Tables have been included in our submission this time. We can assure the reviewer that all data presented is correct and we can fully stand over it.
